# APLA: A SIMPLE ADAPTATION METHOD FOR VISION TRANSFORMERS

## ABSTRACT

Existing adaptation techniques typically require architectural modifications or added parameters, leading to high computational costs and complexity. We introduce Attention Projection Layer Adaptation (APLA), a simple approach to adapt vision transformers (ViTs) without altering the architecture or adding parameters. Through a systematic analysis, we find that the layer immediately after the attention mechanism is crucial for adaptation. By updating only this projection layer, or even just a random subset of this layer's weights, APLA achieves state-of-the-art performance while reducing GPU memory usage by up to 52.63% and training time by up to 43.0%, with no extra cost at inference. Across 46 datasets covering a variety of tasks including scene classification, medical imaging, satellite imaging, and fine-grained classification, APLA consistently outperforms 17 other leading adaptation methods, including full fine-tuning, on classification, segmentation, and detection tasks.

## 1 INTRODUCTION

The primary objective of model adaptation is to enable models to generalize to new tasks with minimal data and computational cost. The most successful approaches accomplish this by injecting new parameters or layers into frozen foundation models Lian et al. (2022); Chen et al. (2022); Jia et al. (2022); Hu et al. (2021). This process often requires complex heuristics – such as gradient sensitivity analyses He et al. (2023); Zhang et al. (2024) and neural architecture searches Zhang et al. (2022)—to determine the optimal locations to inject parameters. Furthermore, the addition of new parameters can introduce significant overhead. Aiming at better efficiency, a handful of methods attempt to adapt the existing structure of the model without adding parameters Zhang et al. (2024); Zaken et al. (2021), but these methods underperform compared to parameter-adding techniques. This raises a critical question: is it possible to achieve competitive adaptation using only a model's existing architecture? We propose that the answer is *yes* – and that the key is to better leverage the model's inherent architecture.

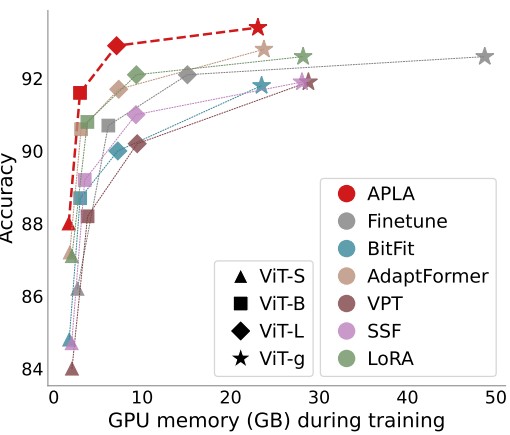

Figure 1: *APLA achieves state-of-the-art for ViT adaption.* It yields better performance for a given GPU memory budget during training compared to full fine-tuning and leading adaptation methods. Similar savings are observed at inference (see Appendix B).

We systematically investigate which existing components of a Vision Transformer (ViT) foundation model are most essential for adaptation in a departure from traditional parameter-adding approaches. Our analysis reveals that the projection layer immediately following the multi-head self-attention (MSA) mechanism plays a uniquely critical role. Then, inspired by low-rank approximation techniques Hu et al. (2021); Jie & Deng (2023), which demonstrate that updates to a full weight matrix can often be effectively represented with lower-dimensional matrices, we explore whether updating the entire projection layer is necessary. We find that *modifying only a random subset of this layer's*

*parameters* is sufficient to maintain – or even surpass – performance, while further reducing computational costs. This result suggests that additional parameters used to learn a low-rank approximation of the updates may be unnecessary, opening the door for simpler, more efficient adaptation strategies.

In this work, we introduce Attention Projection Layer Adaptation (APLA), a novel state-of-the-art approach for efficient adaptation of ViTs that requires no additional parameters. Our key contributions are as follows:

- **Identification of a critical ViT component:** Through systematic experimentation, we identify the projection layer immediately following the attention mechanism as the most essential component for adaptation, offering a targeted approach to ViT tuning essential to APLA which can also improve other adaptation techniques.

- **Low-rank subset update for efficient adaptation:** Building on this insight, we introduce a low-rank adaptation technique that updates only a *random subset* of the projection layer's weights, achieving higher performance with even lower computational costs.

- **Simplified adaptation with no extra parameters:** Our method achieves SOTA results without introducing any new parameters, and eliminates the need for costly heuristics to determine where to inject new parameters or adapt existing ones.

- **Validated across scales and diverse applications:** We validate APLA on 46 datasets across various tasks and model sizes, demonstrating its consistent superiority over 17 adaptation methods. *In most cases, APLA performs better than full fine-tuning* while also achieving up to 52.63% in GPU memory savings and a 43.0% reduction in training time.

Together, these contributions establish APLA as a new standard for efficient and accessible ViT adaptation. Our code for reproducing these experiments will be publicly available upon publication.

## 2 RELATED WORK

Foundation models Bommasani et al. (2021) have transformed computer vision, but as these models grow larger Dehghani et al. (2023); Oquab et al. (2023); Ilharco et al. (2021), their fine-tuning requires high memory and computational resources. The traditional *pretrain, then fine-tune* paradigm Cui et al. (2018); Mustafa et al. (2021); Liu et al. (2021); Zheng et al. (2021) has driven the field for years, but is becoming unfeasible for many applications due to these increasing costs. Recent advancements in model size Dehghani et al. (2023); Oquab et al. (2023); Ilharco et al. (2021) have only increased these challenges, making full fine-tuning unfeasible for many applications.

In response, efficient adaptation methods have emerged, allowing practitioners with fewer resources to leverage large foundation models by introducing only a small set of new parameters, often called parameter-efficient fine-tuning (PEFT). These methods reduce overhead, making large models more deployable in limited-resource settings.

*Adapter-based methods* introduce compact, lightweight modules into specific layers, enabling task-specific adaptation by tuning only the adapter parameters while keeping the base model largely frozen. Originally developed for NLP, Adapters Houlsby et al. (2019) place bottleneck modules sequentially after each multi-head attention and MLP block Vaswani (2017). AdaptFormer Chen et al. (2022) extends this for vision transformers, placing adapters in parallel with the MLP blocks rather than sequentially. More recent methods refine adapter designs for greater efficiency. ARC Dong et al. (2024a) uses a similar bottleneck operation but introduces parameter-sharing. SPT-Adapter He et al. (2023) identifies and adapts only the most impactful layers based on gradient magnitudes. SSF Lian et al. (2022) appends learnable scaling and shifting transformations to modulate features after each ViT layer, while Consolidator Hao et al. (2023) adds grouped connected layers that capture richer information through channel-wise input groups. Adapter-based methods provide flexible, efficient model adaptation with fewer parameters than full fine-tuning. However, they increase inference costs, require careful initialization Steitz & Roth (2024); Houlsby et al. (2019), and their placement often relies on heuristics Chen et al. (2022) or gradient-based selection He et al. (2023), adding computational overhead and potentially leading to suboptimal configurations.

*Low-rank-based methods* leverage the low-rank structure in adaptation updates, enabling efficient adaptation through low-rank matrices. LoRA Hu et al. (2021) pioneered this approach by adding

low-rank matrices alongside original weights in attention blocks. SPT-LoRA He et al. (2023) builds on LoRA by selectively applying low-rank updates to layers with the largest gradient magnitudes. FacT Jie & Deng (2023) and RLRR Dong et al. (2024b) decompose updates into factors, applying these across all ViT layers. While low-rank methods reduce adaptation costs, they insert additional parameters similarly to adapters.

*Prompt-based* methods introduce learnable tokens to guide adaptation without modifying core model parameters. VPT Jia et al. (2022) adds tokens to the input of each transformer block, and $E^2$VPT Han et al. (2023) incorporates auxiliary tokens into attention layers as well. Though prompt-based methods avoid changing internal parameters, they can increase inference costs due to the added tokens. NOAH Zhang et al. (2022) combines prompts with adapters and LoRA modules, using neural architecture search to optimize placement.

*Parameter-selective tuning* is an approach used by a handful of methods most closely related to APLA, that focus on adapting models by tuning only a subset of their existing parameters. GPS Zhang et al. (2024) selects parameters for tuning based on their gradient magnitudes, targeting the most error-inducing parameters during adaptation. BitFit Zaken et al. (2021) takes a simpler approach, updating only the bias parameters. While these methods can be computationally efficient and easy to implement, they face challenges in identifying an optimal subset of parameters, which is reflected in their comparatively poor performance. APLA addresses this by identifying and targeting a critical layer for adaptation in ViTs, achieving state-of-the-art performance.

## 3 METHODS

Inspired by methods that tune a subset of network weights and approaches that use low-rank updates, we ask, "*Can we combine the strengths of both?*" To this end, we identify the most impactful ViT components for adaptation and propose a simple method that updates a low-rank subset of existing weights.

### 3.1 INVESTIGATING ADAPTABILITY OF VIT COMPONENTS

A ViT is composed of multiple learnable components. To identify the most impactful ones for adapting the model to downstream tasks, we first review the different ViT components, grouped by their function (Figure 2).

Starting with an input image $x$, a patchifying stem tiles and reshapes it into $N$ flattened patches. Each patch undergoes a linear transformation in the embedding layer $W_E$ with positional embeddings $\text{Pos}_n, n \in \{1, \ldots, N\}$ added to capture spatial information and a classification token [CLS] appended to create the initial embeddings $z_0$. These embeddings are passed through $L$ transformer blocks, each containing LayerScale (LS) Touvron et al. (2021), LayerNorm (LN) Ba (2016), multi-head self-attention (MSA), and multi-layer perceptron (MLP) modules. The final representation is typically derived from the [CLS] token of the $L$th block, which is then processed by classification head $W_{\text{pred}}$ to produce the prediction $\hat{y}$.

In the MSA block, self-attention is computed for the input tokens using the learnable matrices $W_{Q_i}$, $W_{K_i}$, and $W_{V_i}$, where $i \in \{1, \ldots, h\}$ corresponds to $h$ parallel self-attention heads, allowing each head to learn distinct contextual relationships. The self-attention output for each head is given by:

$$\text{head}_i = \text{softmax}\left(\frac{(z_{in}W_{Q_i})(z_{in}W_{K_i})^T}{\sqrt{d_h}}\right)(z_{in}W_{V_i}) \tag{1}$$

where $d_h$ is the dimensionality of the Query, Key, and Value vectors for each self-attention head. The outputs from each $\text{head}_i$ are concatenated, and a projection layer $W_O$ re-weights the combined features to form the final output of the MSA block.

$$z_{out} = [\text{head}_1; \text{head}_2; \ldots; \text{head}_h]W_O \tag{2}$$

This output is then processed by an MLP block, consisting of two fully connected layers, $W_{FC_1}$ and $W_{FC_2}$, with a non-linearity in between.

To identify the most essential component for adaptation, we conducted an empirical investigation by selectively tuning each of the components described above, one at a time (Figure 2) along with

the final classification head $W_{\text{pred}}$, while keeping the rest of the network frozen. We found that tuning only the projection layer $W_O$—positioned directly after the self-attention operation in the MSA block—yields the best performance, even surpassing full fine-tuning. Details of this study are provided in Section 5.1.

### 3.2 LOW-RANK ADAPTATION THROUGH PARTIAL GRADIENTS

Low-rank adaptation methods leverage the insight that the difference between initial and adapted values of a full-rank matrix can be closely approximated by a low-rank matrix

$$W_{\text{approx}} \approx W_{\text{final}} - W_{\text{init}}, \quad \text{rank}(W_{\text{approx}}) \leq d$$

where $W_{\text{init}}$ and $W_{\text{final}}$ are the layer's learnable matrix before and after adaptation, and $d$ is the full rank. Prior works (*e.g.* Hu et al. (2021); Jie & Deng (2023)) approximate this difference by adding low-rank matrices to ViT layers, with the rank $r := \text{rank}(W_{\text{approx}})$ set as a hyperparameter.

In contrast, we propose a simpler low-rank adaptation by computing gradients on a *randomly selected* subset of columns, which achieves substantial computational savings and retains the benefits of low-rank updates without adding parameters or altering the model's architecture.

Specifically, given a parameter matrix $W \in \mathbb{R}^{d \times d}$, we partition it into a trainable sub-matrix $W_t \in \mathbb{R}^{d \times r}$ where gradients are computed during training, and a frozen sub-matrix $W_f \in \mathbb{R}^{d \times (d-r)}$, which remains unchanged.

$$W_t = W[i, j_m] \qquad m = 1, 2, \ldots, r \quad (3)$$

where the brackets denote indexing with $\{j_1, j_2, \ldots, j_r\} \subseteq \{1, 2, \ldots, d\}$, representing randomly selected trainable column indices, where $r$ controls the rank of the update, reaching full rank when $r = d$.

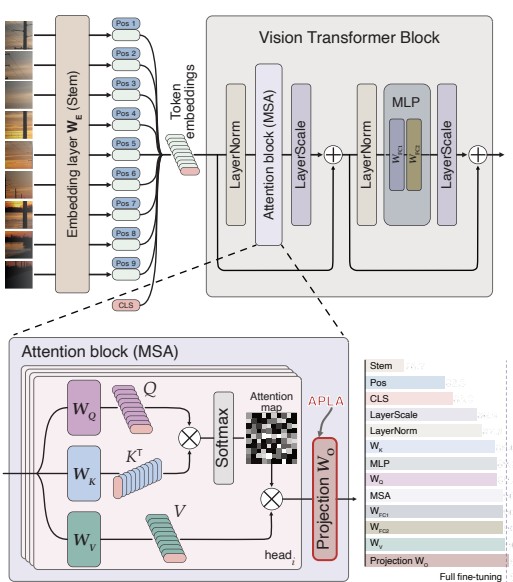

Figure 2: *Investigating adaptation performance of individual ViT components.* We evaluate the adaptation effectiveness of each ViT component in isolation across various downstream tasks, reporting the average performance. Results show that the attention output projection layer ($W_O$), located immediately after the attention mechanism, is the most effective for adaptation. See Section 5.1 and Table 1 for detailed results.

### 3.3 ATTENTION PROJECTION LAYER ADAPTATION (APLA)

Foundation models already encode a rich set of features, and we hypothesize that adapting to new tasks can be achieved by selectively re-weighting these features to fit the target task. Therefore, our approach to efficient model adaptation focuses on identifying impactful layers and computing partial gradients on a randomly selected subset of output features (matrix columns).

The projection layer $W_O$ is an ideal target for adaptation as it plays a central role in re-weighting features from the attention mechanism across all the heads. Therefore, we propose *Attention Projection Layer Adaptation* (APLA), which tunes a randomly selected subset of columns in the $W_O^l$ matrices in each transformer block, while keeping the rest of the ViT backbone frozen. Specifically, we tune only a subset of column vectors of the $W_O^l$ matrices and the final classification head $W_{\text{pred}}$:

$$\mathcal{S}_{\text{APLA}} = \{W_O^1[i, j_m^1], W_O^2[i, j_m^2], \ldots, W_O^L[i, j_m^L], W_{\text{pred}}\} \quad (4)$$

For each transformer block $l \leq L$, we independently sample a distinct subset of column indices $\{j_1^l, j_2^l, \ldots, j_r^l\} \subseteq \{1, 2, \ldots, d\}$ given a global rank hyperparameter $r \leq d$, chosen once at the beginning of training.

APLA is easy to implement, computationally efficient, requires no new parameters, and introduces no additional inference latency, making it highly practical.

## 4 EXPERIMENTAL SETUP

We benchmark APLA against 21 adaptation methods on 46 datasets across 4 tasks, using 3 foundation models. For our main model types, we use Vision Transformers (ViTs) Dosovitskiy (2020) and Swin Transformers Liu et al. (2021) at varying capacities, unless stated otherwise. Below, we provide an overview of our experimental setup. Additional details are available in Appendix A.

**Adaptation methods** We evaluate APLA against various adaptation methods, beginning with traditional approaches: full fine-tuning (FINETUNE) and training only an appended linear layer (LINEAR). We further compare against MLP-$k$ and PARTIAL-$k$. In MLP-$k$, a $k$-layer MLP is appended to the model, and only this block is trained, while PARTIAL-$k$ tunes the last $k$ blocks of the model. We set $k = 3$ for MLP-$k$ and $k = 1$ for PARTIAL-$k$, in line with Jia et al. (2022). To benchmark efficient adaptation, we compare against 17 recent methods: BITFIT Za-

Table 1: *The importance of ViT components for adaptation.* We evaluate how tuning each ViT component in isolation affects performance, while keeping the rest of the model frozen. We report classification performance, with the **best** and second best results highlighted; this notation is used in subsequent tables.

| | Birds | Cars | AID | ISIC | Average |
|---|---|---|---|---|---|
| [CLS] token | 83.9 | 89.9 | 92.9 | 66.4 | 83.3 |
| positional embeddings | 85.1 | 89.6 | 92.9 | 61.4 | 82.3 |
| Embedding layer $W_E$ | 85.7 | 88.2 | 87.9 | 45.1 | 76.7 |
| LayerNorm | 83.7 | 91.5 | 94.6 | 79.0 | 87.2 |
| LayerScale | 85.5 | 91.2 | 93.9 | 75.4 | 86.5 |
| $W_Q$ weight matrix | 85.5 | 91.8 | 94.6 | 85.1 | 89.3 |
| $W_K$ weight matrix | 85.8 | 91.8 | 94.5 | 83.8 | 89.0 |
| $W_V$ weight matrix | 85.8 | 93.2 | 95.3 | 86.9 | 90.3 |
| **$W_O$ weight matrix** | **86.5** | 94.0 | **96.0** | **88.2** | **91.2** |
| MSA block | 85.0 | 93.8 | 94.7 | 86.5 | 90.0 |
| $W_{FC_1}$ weight matrix | 84.7 | 93.5 | 95.0 | 86.9 | 90.0 |
| $W_{FC_2}$ weight matrix | 84.6 | 93.4 | 94.7 | 87.7 | 90.1 |
| MLP block | 82.4 | 93.6 | 94.3 | 86.4 | 89.2 |
| Full Finetuning | 85.2 | **94.4** | 95.4 | 87.7 | 90.7 |

ken et al. (2021) , ADAPTER Houlsby et al. (2019), ADAPTFORMER Chen et al. (2022), VPT-shallow and VPT-deep Jia et al. (2022), E$^2$VPT Han et al. (2023), SSF Lian et al. (2022), LoRA Hu et al. (2021), SPT-adapter and SPT-LoRa He et al. (2023), NOAH Zhang et al. (2022), FACT-TK and FACT-TT Jie & Deng (2023), CONSOLIDATOR Hao et al. (2023) ARC Dong et al. (2024a), GPS Zhang et al. (2024) and RLRR Dong et al. (2024b).

**Datasets and tasks** We benchmark APLA across 46 datasets, covering a diverse set of object categories and tasks. Starting with 21 generic image classification tasks, we cover superordinate object recognition, fine-grained classification, scene recognition, satellite imagery, and medical image analysis using the following datasets: CUB-200-2011 Wah et al. (2011), NABirds Van Horn et al. (2015), Birdsnap Berg

Table 2: *Comparing parameter-selection strategies for APLA.*

| | NABirds | SUN397 | Cal-256 | Cal-101 | Oxf.-Pet | DDSM | Average |
|---|---|---|---|---|---|---|---|
| Largest gradients | **88.1** | 78.3 | 95.3 | 97.4 | 95.9 | **97.2** | 92.0 |
| Largest activations | 87.8 | 78.5 | 95.6 | 97.7 | **96.1** | 96.7 | 92.1 |
| Largest weight magnitude | 87.8 | **78.6** | 95.5 | 97.8 | 96.0 | 97.0 | 92.1 |
| **Random (APLA)** | 88.0 | 78.3 | **95.7** | **98.0** | **96.1** | **97.2** | **92.2** |

et al. (2014), Stanford Dogs Khosla et al. (2011), StanfordCars Krause et al. (2013), Aircraft Maji et al. (2013), Caltech-256 Griffin et al. (2007), Caltech-101 Fei-Fei et al. (2006), CIFAR-100 and CIFAR-10 Krizhevsky et al. (2009), Oxford-III Pet Parkhi et al. (2012), DTD Cimpoi et al. (2014), MIT Indoor Quattoni & Torralba (2009), SUN397 Xiao et al. (2010)), AID Xia et al. (2017), RSSCN7 Cheng et al. (2017), ISIC2019 Tschandl et al. (2018); Codella et al. (2018); Combalia et al. (2019), APTOS2019 Karthik (2019), DDSM Lee et al. (2017), Colorectal Kather et al. (2016), and Pneumonia Kermany et al. (2018).

For semantic & instance segmentation and object detection we use ADE20K Zhou et al. (2019; 2017) and MS COCO Lin et al. (2014). To test in low-data settings, we use VTAB-1k Zhai et al. (2019), a collection of 19 classification tasks with only 1,000 training examples each—representing challenging yet realistic scenarios for model adaptation. For out-of-distribution (OOD) evaluation, we use ImageNet, ImageNet-A Hendrycks et al. (2021b), ImageNet-C Hendrycks & Dietterich (2019), and ImageNet-R Hendrycks et al. (2021a), which introduce various domain shifts, including corruptions, perturbations, and adversarial examples. We use the standard evaluation metric for each dataset.

**Foundation models** To assess the impact of the foundation model type and pretraining strategy, we compare models trained on IMAGENET-21K Deng et al. (2009) using supervised learning, OPENCLIP Ilharco et al. (2021) trained with semi-supervision, and DINOV2 Oquab et al. (2023), pre-trained using self-supervision. We utilize DINOV2 unless stated otherwise.

Table 3: *Main results comparing adaptation methods on image classification* for ViT-B Dosovitskiy (2020) pre-trained with DINOV2 Oquab et al. (2023). The **best** and second best results are highlighted for each task.

| | Fine-grained | | | | | | | General | | | | | | | Scene | | | Satellite | | | Medical | | | | | | Total Average |
|---|---|---|---|---|---|---|---|---|---|---|---|---|---|---|---|---|---|---|---|---|---|---|---|---|---|---|---|
| | CUB-200-2011 | NABirds | Birdsnap | StanfordDogs | StanfordCars | Aircraft | Average | Caltech-256 | Caltech-101 | CIFAR-100 | CIFAR-10 | Oxford-IIIT Pet | DTD | Average | MIT-Indoor | SUN397 | Average | AID | RSSCN7 | Average | ISIC2019 | APTOS2019 | DDSM | Coloretal | Pneumonia | Average | |
| FINETUNE | 88.9 | 85.2 | 78.7 | 86.0 | 94.4 | 87.5 | 86.8 | 93.9 | 97.3 | 92.4 | 98.7 | 94.7 | 81.9 | 93.1 | 87.8 | 75.6 | 81.7 | 95.4 | 73.3 | 84.4 | 87.7 | 90.8 | 95.5 | 97.8 | 99.4 | 94.2 | 89.7 |
| LINEAR | 89.1 | 86.6 | 79.4 | 87.6 | 88.4 | 76.5 | 84.6 | 94.9 | 97.0 | 88.9 | 98.0 | 95.8 | 81.1 | 92.6 | 88.9 | 76.4 | 82.7 | 91.2 | 77.1 | 84.2 | 55.3 | 90.4 | 89.4 | 94.0 | 97.9 | 85.4 | 86.9 |
| MLP | 89.1 | 86.4 | 78.9 | 87.8 | 88.3 | 77.7 | 84.7 | 94.4 | 97.5 | 89.2 | 98.4 | 96.0 | 80.9 | 92.7 | 88.6 | 76.2 | 82.4 | 91.6 | 76.4 | 84.0 | 71.9 | 90.7 | 93.4 | 95.8 | 97.9 | 89.9 | 88.0 |
| PARTIAL | 88.8 | 86.5 | 78.6 | 87.4 | 88.1 | 76.6 | 84.3 | 94.9 | 96.9 | 89.0 | 98.0 | 96.0 | 80.6 | 92.6 | 88.5 | 76.3 | 82.4 | 90.9 | 77.7 | 84.3 | 56.1 | 90.6 | 89.5 | 94.2 | 97.6 | 85.6 | 86.8 |
| BITFIT | 89.4 | 87.9 | 80.7 | 87.8 | 92.5 | 83.2 | 86.9 | 95.2 | 97.6 | 93.1 | 99.3 | 95.7 | 82.2 | 93.9 | 89.3 | 77.7 | 83.5 | 95.2 | 83.5 | 89.4 | 79.0 | 90.1 | 96.4 | 97.2 | 99.1 | 92.4 | 90.1 |
| ADAPTER | 89.6 | 88.4 | 80.0 | 87.8 | 93.5 | 86.1 | 87.6 | 95.0 | 97.6 | 93.5 | 99.3 | 95.9 | 81.8 | 93.9 | 89.5 | 77.9 | 83.7 | 95.0 | 84.2 | 89.6 | 84.3 | 89.6 | 96.1 | 98.0 | 98.5 | 93.3 | 90.6 |
| ADAPTFORMER | 89.7 | 88.4 | 80.5 | 88.1 | 93.1 | 85.4 | 87.5 | 95.6 | 98.1 | 93.2 | 99.3 | 95.9 | 82.8 | 94.2 | 89.9 | 78.1 | 84.0 | 95.4 | 85.3 | 90.4 | 85.6 | 90.8 | 97.3 | 97.4 | 98.6 | 93.9 | 90.9 |
| VPT-SHALLOW | 88.8 | 86.7 | 79.1 | 87.4 | 90.6 | 73.2 | 84.3 | 95.1 | 97.2 | 92.4 | 99.1 | 96.0 | 80.4 | 93.4 | 89.4 | 76.6 | 83.0 | 91.6 | 70.4 | 81.0 | 76.5 | 89.3 | 96.2 | 96.4 | 98.7 | 91.4 | 88.1 |
| VPT-DEEP | 89.1 | 87.3 | 79.9 | 87.2 | 91.5 | 81.7 | 86.1 | 95.3 | 97.8 | 92.7 | 99.1 | 95.7 | 80.5 | 93.5 | 90.0 | 77.0 | 83.5 | 94.4 | 78.0 | 86.2 | 79.6 | 91.0 | 96.2 | 97.6 | 98.8 | 92.6 | 89.5 |
| $E^2$VPT | 88.3 | 86.6 | 79.7 | 87.4 | 91.2 | 81.0 | 85.7 | 94.5 | 96.9 | 92.7 | 99.2 | 95.4 | 79.6 | 93.1 | 88.7 | 76.3 | 82.5 | 93.7 | 72.7 | 83.2 | 80.9 | 90.6 | 96.2 | 96.8 | 98.6 | 92.6 | 88.9 |
| SSF | 89.4 | 88.1 | 80.5 | 87.7 | 92.7 | 83.7 | 87.0 | 95.3 | 97.8 | 93.2 | 99.2 | 95.7 | 82.1 | 93.9 | 88.9 | 77.4 | 83.2 | 95.3 | 82.6 | 89.0 | 80.7 | 90.5 | 96.4 | 97.2 | 99.0 | 92.8 | 90.2 |
| LoRA | 88.7 | 87.5 | 79.3 | 86.3 | 93.4 | 86.4 | 86.9 | 94.3 | 97.1 | 93.0 | 99.0 | 94.0 | 80.4 | 93.0 | 88.5 | 76.4 | 82.5 | 95.4 | 81.4 | 88.4 | 86.5 | 91.1 | 95.1 | 97.4 | 98.9 | 93.8 | 90.0 |
| SPT-ADAPTER | 89.4 | 88.1 | 80.6 | 87.7 | 93.1 | 86.2 | 87.5 | 95.8 | 97.5 | 93.1 | 99.2 | 95.8 | 82.7 | 94.0 | 89.5 | 78.1 | 83.8 | 95.6 | 84.7 | 90.2 | 82.1 | 90.4 | 96.1 | 97.2 | 99.0 | 93.0 | 90.6 |
| SPT-LoRA | 89.2 | 87.9 | 80.6 | 87.5 | 92.8 | 86.3 | 87.4 | 95.8 | 97.7 | 92.6 | 99.2 | 95.7 | 82.3 | 93.9 | 89.9 | 77.7 | 83.8 | 95.4 | 84.2 | 89.8 | 82.2 | 89.3 | 96.2 | 97.4 | 99.1 | 92.8 | 90.4 |
| FACT-TK | 88.8 | 87.8 | 80.5 | 87.5 | 93.0 | 85.4 | 87.2 | 95.3 | 97.6 | 92.8 | 99.2 | 95.5 | 81.5 | 93.7 | 88.9 | 77.4 | 83.2 | 95.4 | 79.9 | 87.7 | 85.1 | 91.5 | 96.3 | 97.2 | 97.9 | 93.6 | 90.2 |
| FACT-TT | 88.8 | 87.6 | 79.7 | 87.1 | 92.9 | 84.3 | 86.7 | 94.9 | 97.5 | 92.4 | 99.2 | 95.5 | 81.7 | 93.5 | 89.4 | 77.1 | 83.3 | 94.5 | 80.6 | 87.6 | 81.5 | 90.9 | 97.0 | 97.4 | 98.6 | 93.1 | 89.9 |
| CONSOLIDATOR | 89.7 | 87.4 | 81.5 | 87.4 | 91.2 | 83.4 | 87.0 | 94.5 | 97.3 | 92.7 | 99.0 | 95.8 | 81.8 | 93.5 | 89.5 | 77.0 | 83.3 | 94.6 | 78.0 | 86.3 | 81.9 | 91.1 | 96.9 | 96.2 | 99.4 | 93.1 | 89.9 |
| ARC | 89.4 | 88.2 | 80.7 | 88.1 | 92.6 | 84.1 | 87.2 | 95.8 | 97.8 | 92.9 | 99.2 | 96.0 | 82.8 | 94.1 | 89.8 | 78.2 | 84.0 | 95.6 | 86.2 | 90.9 | 82.5 | 90.7 | 97.3 | 97.0 | 98.8 | 93.3 | 90.7 |
| GPS | 89.1 | 86.4 | 80.7 | 86.6 | 94.7 | 85.5 | 87.2 | 94.8 | 97.6 | 94.0 | 99.3 | 94.4 | 77.9 | 93.0 | 88.4 | 76.9 | 82.7 | 94.9 | 61.6 | 78.3 | 87.7 | 90.8 | 96.7 | 97.4 | 99.3 | 94.4 | 89.3 |
| RLRR | 88.9 | 87.9 | 80.8 | 87.6 | 92.4 | 83.7 | 86.9 | 95.2 | 97.4 | 93.1 | 99.2 | 95.8 | 82.0 | 93.8 | 89.5 | 77.6 | 83.6 | 95.0 | 81.2 | 88.1 | 81.7 | 90.4 | 96.3 | 96.6 | 98.6 | 92.7 | 90.0 |
| **APLA** | 89.6 | 88.0 | 81.9 | 88.5 | 94.0 | 86.7 | 88.1 | 95.7 | 98.0 | 93.4 | 99.3 | 96.1 | 83.0 | 94.3 | 90.4 | 78.3 | 84.4 | 96.0 | 86.2 | 91.1 | 88.2 | 92.1 | 97.2 | 98.4 | 99.5 | 95.1 | 91.5 |

**Implementation details** To ensure fair comparison, we closely follow the implementations in Jia et al. (2022); Dong et al. (2024a;b); Jie & Deng (2023); Han et al. (2023); Chen et al. (2022); Hu et al. (2021). For each dataset, we use the official protocol and standard train/val/test splits when available Zhai et al. (2019) or the splits provided by Jia et al. (2022). Models are trained with AdamW Loshchilov & Hutter (2017) for 100 epochs, using a cosine decay learning rate schedule with a 10-epoch warm-up. Hyperparameters are selected via grid search on the validation set. Additional details regarding the experimental setup can be found in Appendix A.

## 5 EXPERIMENTS AND RESULTS

We begin by evaluating the choice of APLA 's core components, focusing first on identifying the most crucial layer for adaptation—the projection layer $W_O$—and how to select which parameters to tune within it. We then benchmark APLA against several other methods on standard classification, detection, and segmentation tasks, including low-data settings and out-of-distribution datasets. Additionally, we assess APLA across various model capacities, architectures, and foundation models. Finally, we evaluate its computational efficiency and explore how other adaptation techniques can benefit from our findings on the importance of the projection layer $W_O$.

### 5.1 CHOOSING APLA'S COMPONENTS

**Identifying which component to tune** Previous work suggests that certain types of layers play a larger role in transfer learning Yosinski et al. (2014); Sharif Razavian et al. (2014); Neyshabur et al. (2020); Matsoukas et al. (2022); Konuk et al. (2024). To identify the optimal components for APLA, we systematically investigate the effect of tuning different ViT components individually, keeping all other layers frozen.

Table 1 and Figure 2 present results across four mainstream datasets representing diverse domains, tasks, and data availabilities. We find that tuning the projection layer $W_O$ yields the best performance, even outperforming full-network fine-tuning. The clear advantage of $W_O$ over other ViT components leads us to adopt it as the default layer to tune in APLA.

**Identifying which parameters to tune** Motivated by the effectiveness of low-rank adaptation methods, we explore strategies to economize APLA by selecting specific parameters in $W_O$ for adaptation. We evaluate various strategies, including selecting columns with the largest gradients, activations, and weight magnitudes, as well as selecting the columns randomly.

We report our findings in Table 2. Surprisingly, tuning a random subset of columns in $W_O$ performs on par with more sophisticated selection methods. This suggests that, within the projection layer $W_O$, the specific choice of tunable columns is less critical. For APLA we choose the random selection approach because it may better utilize feature redundancy across attention heads and achieves a slight advantage in performance without added computational cost.

## 5.2 BENCHMARKING APLA

**Mainstream classification tasks** We benchmark APLA against other methods across 21 diverse image classification tasks, including superordinate classification, fine-grained classification, scene recognition, satellite imagery, and medical image analysis. Table 3 presents the results. On average, APLA outperforms all other methods, showing a 0.6% improvement over the second-best method, ADAPTFORMER. It ranks as the top or second-best method on 18 out of 21 datasets, consistently demonstrating strong performance across different classification tasks.

**APLA in low-data settings** Foundation model adaptation is especially valuable in low-data scenarios, where reusing pretrained features is essential, as randomly initialized models tend to underperform Dosovitskiy (2020); Kolesnikov et al. (2020); Matsoukas et al. (2023). We evaluate APLA in this regime using the VTAB-1k benchmark, which includes 19 datasets, each with only 1,000 samples. We use ViT-B and Swin-B pretrained on IMAGENET-21K and report average performance across three domains: natural, specialized, and structured. As shown in Table 4, APLA outperforms all other methods, achieving at least a 1% improvement across these domains. Visualizations Van der Maaten & Hinton (2008) of the [CLS] embeddings of different adaptation methods from these experiments are provided in Figure 8 in the Appendix.

Table 4: *Low-data settings.* We benchmark on VTAB-1k which contains 19 low-data tasks grouped across three domains.

| | ViT-B | | | | Swin-B | | | |
|---|---|---|---|---|---|---|---|---|
| | Natrl. | Spec. | Struc. | Average | Natrl. | Spec. | Struc. | Average |
| FINETUNE | 75.9 | 83.4 | 47.6 | 69.0 | 79.1 | 86.2 | 59.7 | 75.0 |
| LINEAR | 68.9 | 77.2 | 26.9 | 57.7 | 73.5 | 80.8 | 33.5 | 62.6 |
| MLP | 67.8 | 72.8 | 30.6 | 57.1 | 73.6 | 75.2 | 35.7 | 61.5 |
| PARTIAL | 69.4 | 78.5 | 34.2 | 60.7 | 73.1 | 81.7 | 35.0 | 63.3 |
| BITFIT | 73.3 | 78.3 | 44.1 | 65.2 | 74.2 | 80.1 | 42.4 | 65.6 |
| ADAPTER | 79.0 | 84.1 | 58.5 | 73.9 | 81.7 | 87.3 | 61.2 | 76.7 |
| ADAPTFORMER | 80.6 | 85.4 | 58.8 | 74.9 | – | – | – | – |
| VPT-SHALLOW | 76.8 | 79.7 | 47.0 | 67.8 | 79.9 | 82.5 | 37.8 | 66.7 |
| VPT-DEEP | 78.5 | 82.4 | 55.0 | 72.0 | 76.8 | 84.5 | 53.4 | 71.6 |
| E²VPT | 80.0 | 84.4 | 57.4 | 73.9 | 83.3 | 85.0 | 57.4 | 75.2 |
| SSF | 81.6 | 86.6 | 59.0 | 75.7 | – | – | – | – |
| LoRA | 79.5 | 84.6 | 60.5 | 74.8 | 81.7 | 87.2 | 60.1 | 76.3 |
| SPT-ADAPTER | 82.0 | 85.8 | 61.4 | 76.4 | 83.0 | 87.3 | 62.1 | 77.5 |
| SPT-LoRA | 81.9 | 85.9 | 61.3 | 76.4 | 83.1 | 87.4 | 60.4 | 77.2 |
| NOAH | 80.2 | 84.9 | 61.3 | 75.5 | – | – | – | – |
| CONSOLIDATOR | 82.4 | 86.3 | 60.9 | 76.5 | – | – | – | – |
| FACT-TK | 80.6 | 85.3 | 60.7 | 75.5 | – | – | – | – |
| FACT-TT | 80.6 | 85.0 | 60.5 | 75.3 | 83.1 | 86.9 | 62.1 | 77.4 |
| ARC | 81.8 | 87.0 | 61.4 | 76.7 | 79.0 | 86.6 | 59.9 | 75.2 |
| RLRR | 83.7 | 87.3 | 61.8 | 77.6 | 81.3 | 86.7 | 59.0 | 75.7 |
| GPS | 83.7 | 86.8 | 61.9 | 77.5 | – | – | – | – |
| **APLA** | **84.6** | **88.5** | **62.7** | **78.6** | **84.4** | **87.8** | **65.9** | **79.4** |

**Out-of-distribution robustness** While APLA has shown strong performance across various settings, its robustness under domain shifts and adversarial examples remains to be assessed. Using a foundation model pre-trained on ImageNet-21K, tuned on ImageNet-1K with various adaptation methods, we evaluate on ImageNet-A Hendrycks et al. (2021b), ImageNet-R Hendrycks et al. (2021a), and ImageNet-C Hendrycks & Dietterich (2019).

As shown in Table 5, APLA outperforms other methods overall, achieving an 8.6% improvement in mean corruption error (mCE) on ImageNet-C. Notably, APLA and most other adaptation methods outperform full fine-tuning across all OOD datasets, highlighting the potential of efficient adaptation methods for OOD tasks.

Table 5: *OOD robustness.* We assess robustness to OOD data by adapting on ImageNet-1K and testing on ImageNet-A, ImageNet-R, and ImageNet-C.

| | ImageNet-1K Acc. (↑) | ImageNet-A Acc. (↑) | ImageNet-R Acc. (↑) | ImageNet-C mCE (↓) |
|---|---|---|---|---|
| FINETUNE | 83.6 | 34.5 | 51.3 | 46.5 |
| LINEAR | 82.0 | 33.9 | 52.9 | 46.9 |
| ADAPTER | 82.7 | 42.2 | 54.1 | 42.7 |
| BITFIT | 82.7 | 42.1 | 55.9 | 41.9 |
| VPT-SHALLOW | 82.1 | 30.9 | 53.7 | 46.9 |
| VPT-DEEP | 82.5 | 39.1 | 53.5 | 43.1 |
| SSF | 83.1 | 45.9 | 56.8 | 41.5 |
| GPS | 83.9 | 46.1 | **57.0** | 42.0 |
| **APLA** | **84.0** | **46.9** | 55.5 | **32.9** |

**Segmentation & Detection Tasks** We evaluate APLA on semantic segmentation, object detection, and instance segmentation. For semantic segmentation, we use SETR-PUP Zheng et al. (2021) with a ViT-Large backbone pre-trained on IMAGENET-21K, reporting mean Intersection over Union (mIoU) for single-scale (SS) and multi-scale (MS) evaluations on ADE20K, as in Jia et al. (2022); He et al. (2023).

Table 7: *Impact of pre-training strategy.* ViT-B pre-trained with IMAGENET-21K and OPENCLIP, then adapted to various tasks.

| | IMAGENET-21K | | | | | OPENCLIP | | | | |
|---|---|---|---|---|---|---|---|---|---|---|
| | Birds | Cars | AID | ISIC | Average | Birds | Cars | AID | ISIC | Average |
| FINETUNE | 82.7 | 84.5 | 91.7 | 84.0 | 85.7 | 79.0 | 94.7 | 95.6 | 84.9 | 88.6 |
| LINEAR | 75.9 | 51.3 | 81.0 | 51.2 | 64.9 | 73.7 | 94.5 | 95.0 | 54.2 | 79.4 |
| MLP | 77.3 | 54.9 | 81.2 | 61.7 | 68.8 | 73.6 | 93.8 | 95.0 | 68.7 | 82.8 |
| PARTIAL | 77.8 | 66.2 | 81.1 | 46.6 | 67.9 | 73.8 | 94.4 | 95.2 | 54.9 | 79.6 |
| BITFIT | 84.2 | 79.4 | 90.5 | 73.0 | 81.8 | 79.2 | 95.0 | 95.8 | 72.7 | 85.7 |
| ADAPTER | 84.3 | 68.6 | 90.0 | 80.6 | 80.9 | 79.2 | 95.0 | 95.2 | 83.9 | 88.3 |
| ADAPTFORMER | 78.8 | 83.1 | 90.1 | 81.2 | 83.3 | 80.0 | 95.0 | 95.5 | 82.4 | 88.2 |
| VPT-SHALLOW | 78.8 | 68.7 | 85.9 | 65.0 | 74.6 | 73.8 | 94.7 | 95.1 | 54.6 | 79.6 |
| VPT-DEEP | 84.2 | 83.6 | 89.0 | 74.8 | 82.9 | 73.6 | 94.5 | 95.1 | 54.6 | 79.5 |
| E²VPT | 84.6 | 82.8 | 88.4 | 78.6 | 83.6 | 77.7 | 95.1 | 95.9 | 73.6 | 85.6 |
| SSF | 85.7 | 89.2 | 90.9 | 78.4 | 86.1 | 79.9 | 94.9 | 95.7 | 76.1 | 86.7 |
| LoRA | 85.6 | 83.2 | 91.0 | 83.5 | 85.8 | 79.0 | 94.1 | 95.0 | 83.2 | 87.8 |
| SPT-ADAPTER | 83.3 | 86.2 | 90.8 | 75.7 | 84.0 | 76.0 | 94.8 | 95.4 | 76.4 | 85.7 |
| SPT-LoRA | 83.4 | 87.3 | 90.0 | 76.2 | 84.2 | 75.9 | 94.9 | 95.1 | 77.4 | 85.8 |
| FacT-TK | 80.3 | 84.0 | 91.1 | 79.3 | 83.7 | 79.2 | 94.9 | 95.0 | 80.7 | 87.5 |
| FacT-TT | 79.2 | 82.4 | 90.9 | 77.7 | 82.6 | 78.5 | 94.9 | 95.4 | 77.2 | 86.5 |
| ARC | 85.7 | 89.5 | 90.8 | 79.2 | 86.3 | 79.5 | 95.1 | 95.1 | 79.0 | 87.2 |
| RLRR | 85.3 | 90.4 | 91.1 | 77.5 | 86.1 | 80.0 | 94.9 | 95.8 | 79.5 | 87.6 |
| APLA | 85.2 | 90.5 | 94.3 | 84.9 | 88.7 | 79.1 | 95.2 | 95.9 | 85.5 | 88.9 |

Table 8: *Impact of model capacity.* Results are averaged across NABirds, StanfordCars, AID, and ISIC2019 using DINOV2 models. Detailed results are in Appendix B.2.

| | ViT-S | ViT-B | ViT-L | ViT-g |
|---|---|---|---|---|
| FINETUNE | 86.2 | 90.7 | 92.1 | 92.6 |
| LINEAR | 76.1 | 80.4 | 83.6 | 85.5 |
| MLP | 80.9 | 84.6 | 86.2 | 87.8 |
| PARTIAL | 75.8 | 80.4 | 83.6 | 85.2 |
| BITFIT | 84.8 | 88.7 | 90.0 | 91.8 |
| ADAPTER | 87.9 | 90.3 | 91.9 | 92.3 |
| ADAPTFORMER | 87.2 | 90.6 | 91.7 | 92.8 |
| VPT-SHALLOW | 82.8 | 86.4 | 88.3 | 88.9 |
| VPT-DEEP | 84.0 | 88.2 | 90.2 | 91.9 |
| E²VPT | 84.2 | 88.1 | 90.9 | 91.7 |
| SSF | 84.7 | 89.2 | 91.0 | 91.9 |
| LoRA | 87.1 | 90.8 | 92.1 | 92.6 |
| SPT-ADAPTER | 87.2 | 89.7 | 90.0 | 89.5 |
| SPT-LoRA | 87.6 | 89.6 | 90.5 | 89.4 |
| FacT-TK | 86.9 | 90.3 | 91.5 | 92.0 |
| FacT-TT | 85.4 | 89.1 | 91.3 | 91.6 |
| ARC | 86.0 | 89.7 | 91.1 | 91.9 |
| RLRR | 84.4 | 89.3 | 91.8 | 92.3 |
| APLA | 88.0 | 91.6 | 92.9 | 93.4 |

For object detection and instance segmentation, we use Mask R-CNN He et al. (2017) with a Swin-Tiny backbone pre-trained on IM-AGENET-1K, following Lian et al. (2022); Liu et al. (2021), and report mean Average Precision (AP) for bounding box ($AP^{bb}$) and mask ($AP^m$) predictions on MS COCO Lin et al. (2014). Additional details are in Appendix A. As shown in Table 6, APLA surpasses all other adaptation methods, with particularly strong results for semantic segmentation.

**Different foundation model types and capacities.** We evaluate APLA 's versatility across foundation model training strategies, capacities, and architectures, using both supervised IMA-GENET-21K and semi-supervised OPENCLIP ViT-B models, as well as Swin transformers Liu et al. (2021). To assess scalability, we test ViT models of varying sizes (ViT-S, ViT-B, ViT-L, and ViT-g). As shown in Tables 4, 7, and 8, APLA consistently outperforms other methods regardless of pretraining, model size, or architecture, maintaining strong performance.

Table 6: *Segmentation and detection.* Results for ADE20K semantic segmentation (SETR-PUP Zheng et al. (2021) with a ViT-Large backbone) and COCO object detection & instance segmentation (Mask R-CNN He et al. (2017) with a Swin-Tiny backbone).

| | ADE20K | | COCO | | | | | |
|---|---|---|---|---|---|---|---|---|
| | mIoU (SS) | mIoU (MS) | $AP^{bb}$ | $AP^{bb}_{50}$ | $AP^{bb}_{75}$ | $AP^m$ | $AP^m_{50}$ | $AP^m_{75}$ |
| BITFIT | 43.4 | 45.3 | 33.7 | 57.8 | 35.0 | 32.7 | 54.7 | 33.9 |
| VPT-DEEP | 42.1 | 44.1 | 33.8 | 57.6 | 35.3 | 32.5 | 54.5 | 33.9 |
| SSF | 45.6 | 47.4 | 34.9 | 58.9 | 36.1 | 33.5 | 55.8 | 34.7 |
| LoRA | 43.9 | 45.9 | 37.1 | 60.9 | 39.5 | 35.2 | 57.7 | 37.2 |
| ADAPTER | 44.4 | 46.6 | 37.6 | 61.1 | 40.2 | 35.6 | 58.2 | 37.8 |
| ADAPTFORMER | 44.3 | 46.2 | 35.1 | 59.1 | 36.9 | 33.8 | 56.0 | 35.6 |
| SPT-ADAPTER | 45.2 | 47.2 | – | – | – | – | – | – |
| SPT-LoRA | 45.4 | 47.5 | – | – | – | – | – | – |
| APLA | 48.3 | 49.5 | 38.1 | 61.8 | 40.9 | 35.9 | 58.7 | 37.9 |

**Applying other adaptation methods on $W_O$** In Section 5.1 we showed that $W_O$ is the most impactful component to adapt—surpassing full fine-tuning—and in Section 5.2 we show that, when targeted in APLA, it outperforms other adaptation methods. We now explore what happens if other adaptation methods are applied to $W_O$. Do they improve performance when targeted to this layer? Does the low-rank adaptation strategy we propose for APLA prevail against other adaptation methods that target the same layer? Using ViT-B pretrained with DINOV2, we apply LoRA Hu et al. (2021), FacT Jie & Deng (2023), and ADAPTFORMER Chen et al. (2022) on $W_O$ and compare on Birds, Cars, AID, ISIC, and VTAB-1k.

Table 10 in the Appendix shows that applying other adaptation methods to $W_O$ generally improves performance, solidifying the importance of the $W_O$ layer. *Critically, APLA still outperforms other leading approaches when they are applied to $W_O$, suggesting there is an advantage to our simple low-rank adaptation strategy using random partial gradients.*

**Computational cost** We analyze the computational costs of adaptation methods by measuring GPU memory footprint, parameter count, and throughput during training and inference in Appendix B. APLA demonstrates significant efficiency improvements, reducing memory usage and boosting training throughput, with no extra inference cost. Figure 1 further illustrates that APLA's memory savings increase with model size, even surpassing BitFit, which tunes only bias parameters. Figure

4 in the Appendix reports parameter count, showing that APLA appears more costly than many methods according to this metric. However, as noted by prior work Dehghani et al. (2021); Cai et al. (2020), parameter count is misleading in assessing computational efficiency. In practice, *APLA remains the most efficient method*, consistently outperforming others in GPU memory usage and throughput during both training and inference. Further, Appendix B.4 shows that APLA maintains near-constant efficiency as the rank $r$ increases, unlike other low-rank methods.

## 6 DISCUSSION

*APLA establishes a new state-of-the-art for efficient model adaptation* across a wide range of classification, segmentation, and detection tasks, showing resilience in low-data and out-of-distribution scenarios. It achieved top performance across model types, capacities, and pretraining methods – a level of versatility that no other adaptation method maintained across such varied conditions. Moreover, APLA excels in computational efficiency, significantly reducing memory and processing requirements.

*APLA offers several practical advantages* in addition to its exceptional performance. It is simple to implement, requiring no architectural changes or added parameters which may be sensitive to initialization. It eliminates the need to search for which parameters to update. APLA's simplicity makes it easy to work with. APLA also supports flexible layer adaptation, allowing partial or full layer updates depending on the need, all with minimal computational overhead. Finally, as demonstrated in Section 5.2, APLA's core insights can be used to enhance other adaptation methods, *e.g.* applying ADAPTFORMER solely on $W_O$ gives a 1% boost in performance.

*What makes APLA so effective?* Although we don't have a definitive answer, we offer two possible explanations. One is the targeting of $W_O$. Foundation models encode a rich set of features robust to various tasks. However, each task benefits from a unique composition of these features, making feature re-weighting essential. This is precisely the role of $W_O$, which re-weights the contribution of features across all attention heads. Figure 2 reveals that other top-performing ViT components serve similar functions: $W_V$ re-weights the attention output within each head, while $W_O$ operates across all heads. Given this, one might expect $W_{FC_1}$ and $W_{FC_2}$ in the MLP block to play a more critical role, but they are positioned further downstream, with $W_O$ and normalization layers modifying the features before they reach the MLP block.

A second explanation for APLA's effectiveness lies in the simplicity of its low-rank adaptation using randomly selected gradient updates. Other approaches to use heuristics to select parameters, *e.g.* based on large weights, activations, or gradients may be suboptimal for foundation models, which are highly over-parameterized and exhibit feature redundancy. Selection based on large gradients or weights may not capture the most relevant features, could bias adaptation toward redundant or overly specific features, and lead to overfitting. By re-weighting a broader range of features, random selection makes APLA equally or even more effective in contexts of high feature redundancy, as shown in Table 2.

**Limitations & future work**   While our experiments are extensive, certain aspects remain unexplored. Our study focuses on identifying the single most important ViT component for adaptation rather than multiple components. An exhaustive search would be computationally prohibitive, and a constrained search, resembling a NAS, may yield undesirable task-specific combinations Zhang et al. (2022). We also did not examine how the choice of $r$ might vary with data availability or information density; richer data may support a larger $r$ and enhance adaptability. Lastly, APLA 's susceptibility to catastrophic forgetting remains untested –unlike adapter-based methods, which can be stored separately, APLA directly modifies the foundation model, potentially impacting retention of prior knowledge.

## 7 CONCLUSION

We introduced APLA, a simple yet effective method for adapting ViTs by tuning only a randomly selected subset of projection layer columns. Extensive experiments show that APLA achieves state-of-the-art performance while reducing computational costs, making it highly practical. Our results highlight that in over-parameterized models, efficiency doesn't require added complexity – a simple targeted re-weighting of existing features can be even more powerful.

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

# Appendix

We provide additional experimental details and results.

- A includes additional experimental details.
  - In Section A.1 we provide implementation details for classification tasks.
  - In A.2 we provide experimental details for semantic segmentation tasks.
  - In A.3 we provide experimental details for object detection and instance segmentation tasks.
- Section B includes additional experimental results.
  - In B.1, we show the result of applying other low-rank methods on $W_O$.
  - In B.2 we report additional results as the model's capacity increases, including both classification performance and computational cost.
  - In B.3 we report additional results on the computational costs of APLA and other adaptation methods during training and inference.
  - In B.4 We examine the effect of rank $r$ on computational requirements for different low-rank methods when applied solely to $W_O$ during training.
  - In B.5 we investigate the impact of applying APLA to an increasing number of ViT blocks.
  - In B.6 we discuss the best $r$ values for APLA.
  - In B.7 we visualize and discuss the quality of output features produced by different adaptation methods.

## A  ADDITIONAL EXPERIMENTAL DETAILS

### A.1  IMAGE CLASSIFICATION

To ensure fair comparison against other adaptation methods, we closely follow the implementations in Jia et al. (2022); Dong et al. (2024a;b); Jie & Deng (2023); Han et al. (2023); Chen et al. (2022); Hu et al. (2021). For each dataset in the general classification tasks, either the official train/val/test splits were used, or we used the splits from Jia et al. (2022). For VTAB, the train/val/test splits are provided. Similar to Jia et al. (2022), we adopt standard image augmentations, including random resized crop to $224 \times 224$, random horizontal flip, and normalization with mean and std. For VTAB, we resize all images to $224 \times 224$. All models were developed in PyTorch Paszke et al. (2019) and trained on Nvidia A100 GPUs using AdamW Loshchilov & Hutter (2017) optimizer. Unless stated otherwise, models were trained for 100 epochs using a cosine decay learning rate schedule with a 10-epoch warm-up, following previous works Jia et al. (2022); Dong et al. (2024a;b); Jie & Deng (2023); Han et al. (2023). We perform grid-search to determine the hyper-parameters using the validation set of each dataset. We also perform a grid search to determine the appropriate $r$ value in APLA for each dataset. For ViT-S, we search over $r \in \{8, 16, 128, 256, 384\}$. For ViT-B, we search over $r \in \{8, 16, 128, 512, 768\}$. For ViT-L, we search over $r \in \{8, 16, 128, 512, 1024\}$. For ViT-g, we search over $r \in \{8, 16, 128, 1024, 1536\}$.

### A.2  SEMANTIC SEGMENTATION

For semantic segmentation we follow Jia et al. (2022); He et al. (2023) and conduct experiments on the ADE20K dataset Zhou et al. (2019; 2017) using the SETR-PUP framework Zheng et al. (2021) with a ViT-Large Dosovitskiy (2020) model pre-trained on IMAGENET-21K Deng et al. (2009). We report mean Intersection over Union (mIoU) scores for both single-scale (SS) and multi-scale (MS), following Jia et al. (2022); He et al. (2023). Our implementation uses the *mmsegmentation* Contributors (2020) library. We merely apply APLA on the default models of the library. All training configurations are kept unchanged.

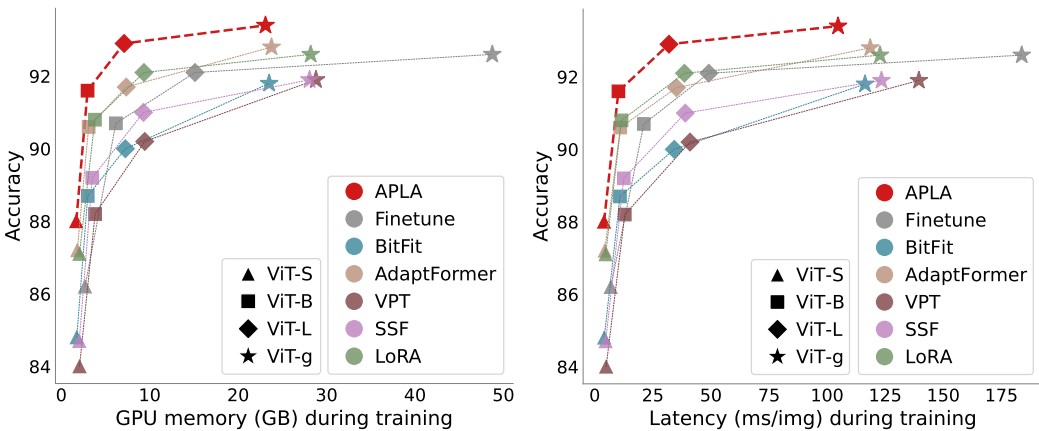

Figure 3: *Performance vs. compute cost.* We compare each method's performance against GPU memory (left) and latency (right) during training across different model capacities.

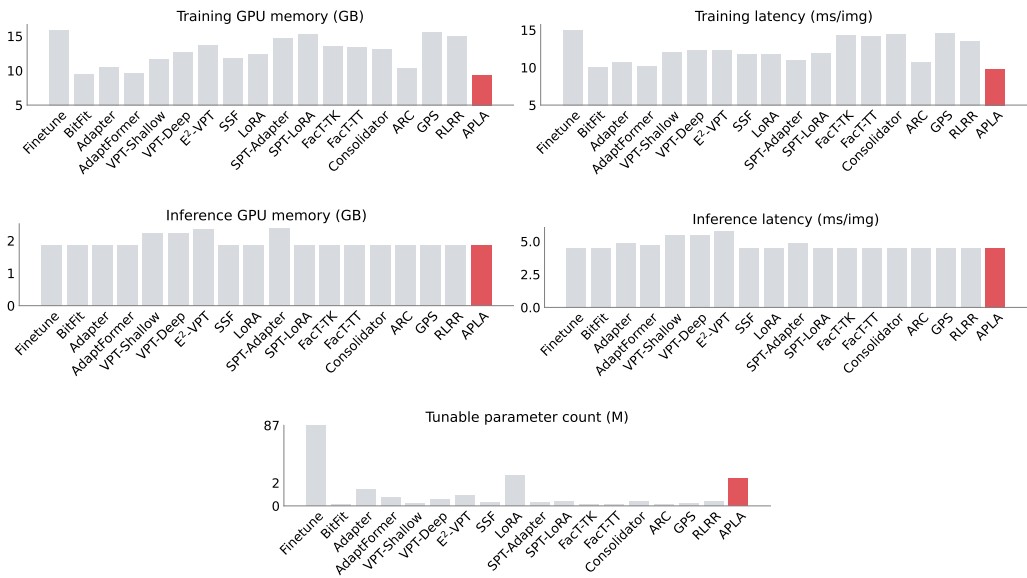

Figure 4: *Computational costs.* We report memory footprint and latency of various adaptation methods during training (top) and inference (middle) for ViT-B with a batch size of 64. Additionally, we provide the number of tunable parameter count for each method (bottom), averaged across all the datasets.

## A.3 OBJECT DETECTION & INSTANCE SEGMENTATION

For object detection and instance segmentation tasks we follow Lian et al. (2022); Liu et al. (2021) and conduct experiments on the MS COCO dataset Lin et al. (2014) using the Mask R-CNN framework He et al. (2017) with a Swin-Tiny Liu et al. (2021) model pre-trained on IMAGENET-1K Deng et al. (2009). We report mean Average Precision (AP) for both bounding boxes ($AP^{bb}$) and masks ($AP^m$) across multiple IoU thresholds and individual thresholds, following Lin et al. (2014); Lian et al. (2022); Liu et al. (2021). Our implementation uses the *mmdetection* Chen et al. (2019) library. We merely apply APLA on the default models of the library. All training configurations are kept unchanged.

Table 9: *Classification performance across different model sizes.* The **best** and second best results are highlighted.

| | ViT-S | | | | | ViT-B | | | | | ViT-L | | | | | ViT-g | | | | |
|---|---|---|---|---|---|---|---|---|---|---|---|---|---|---|---|---|---|---|---|---|
| | Birds | Cars | AID | ISIC | Average | Birds | Cars | AID | ISIC | Average | Birds | Cars | AID | ISIC | Average | Birds | Cars | AID | ISIC | Average |
| FINETUNE | 77.5 | **91.9** | 91.7 | 83.8 | 86.2 | 85.2 | **94.4** | 95.4 | 87.7 | 90.7 | 88.2 | 94.9 | 95.9 | **89.2** | 92.1 | 90.0 | 95.2 | 96.5 | 88.8 | 92.6 |
| LINEAR | 81.3 | 83.1 | 88.5 | 51.5 | 76.1 | 86.6 | 88.4 | 91.2 | 55.3 | 80.4 | 89.1 | 89.8 | 93.4 | 62.1 | 83.6 | 90.2 | 91.0 | 93.6 | 67.3 | 85.5 |
| MLP | 80.6 | 83.1 | 88.8 | 71.1 | 80.9 | 86.4 | 88.3 | 91.6 | 71.9 | 84.6 | 88.9 | 89.9 | 92.9 | 73.0 | 86.2 | 89.9 | 91.0 | 93.1 | 77.1 | 87.8 |
| PARTIAL | 81.1 | 83.1 | 88.3 | 50.6 | 75.8 | 86.5 | 88.1 | 90.9 | 56.1 | 85.2 | 89.1 | 89.8 | 93.3 | 62.0 | 83.6 | 90.3 | 91.0 | 93.6 | 65.7 | 85.2 |
| BITFIT | 83.1 | 89.7 | 93.1 | 73.2 | 84.8 | 87.9 | 92.5 | 95.2 | 79.0 | 88.7 | 90.4 | 93.8 | 95.8 | 80.1 | 90.0 | 90.8 | 94.5 | 95.9 | 85.8 | 91.8 |
| ADAPTER | 83.2 | 91.3 | 93.3 | 83.8 | 87.9 | **88.4** | 93.5 | 95.0 | 84.3 | 90.3 | 90.4 | 94.8 | 95.8 | 86.4 | 91.9 | 91.1 | 95.1 | 96.3 | 86.8 | 92.3 |
| ADAPTFORMER | 83.6 | 90.6 | 92.8 | 81.9 | 87.2 | **88.4** | 93.1 | 95.4 | 85.6 | 90.6 | **90.8** | 94.2 | 95.8 | 86.0 | 91.7 | 91.5 | 94.9 | 96.0 | 88.9 | 92.8 |
| VPT-SHALLOW | 81.7 | 86.3 | 91.3 | 72.0 | 82.8 | 86.7 | 90.6 | 91.6 | 76.5 | 86.4 | 89.0 | 91.6 | 93.0 | 79.7 | 88.3 | 89.8 | 92.1 | 95.1 | 78.7 | 88.9 |
| VPT-DEEP | 80.1 | 86.8 | 92.5 | 76.7 | 84.0 | 87.3 | 91.5 | 94.4 | 79.6 | 88.2 | 89.1 | 93.4 | 95.7 | 82.6 | 90.2 | 91.1 | 94.5 | 96.2 | 85.9 | 91.9 |
| E²VPT | 80.0 | 87.6 | 91.9 | 77.3 | 84.2 | 86.6 | 91.2 | 93.7 | 80.9 | 88.1 | 89.5 | 93.8 | 95.8 | 84.3 | 90.9 | 91.2 | 94.5 | 95.9 | 85.3 | 91.7 |
| SSF | 81.7 | 89.5 | 92.7 | 74.9 | 84.7 | 88.1 | 92.7 | 95.3 | 80.7 | 89.2 | 90.6 | 94.0 | 95.9 | 83.5 | 91.0 | 91.0 | 94.5 | 95.9 | 86.3 | 91.9 |
| LoRA | 80.8 | 91.0 | 93.3 | 83.2 | 87.1 | 87.9 | 93.4 | 95.4 | 86.5 | 90.8 | 89.9 | 94.8 | 95.9 | 87.9 | 92.1 | 89.8 | 94.8 | 96.7 | 88.9 | 92.6 |
| SPT-ADAPTER | 83.0 | 91.0 | 93.3 | 81.4 | 87.2 | 88.1 | 93.1 | 95.6 | 82.1 | 89.7 | 90.6 | 93.4 | 95.6 | 80.5 | 90.0 | 90.7 | 93.2 | 95.0 | 79.0 | 89.5 |
| SPT-LORA | 82.8 | 91.1 | 93.2 | 83.2 | 87.6 | 87.9 | 92.8 | 95.4 | 82.2 | 89.6 | 89.6 | 93.8 | 95.5 | 82.9 | 90.5 | 90.3 | 93.5 | 95.2 | 78.5 | 89.4 |
| FACT-TK | 82.5 | 90.6 | 92.9 | 81.7 | 86.9 | 87.8 | 93.0 | 95.4 | 81.5 | 90.3 | 90.6 | 94.5 | 96.3 | 84.4 | 91.5 | 91.6 | 95.1 | 96.1 | 85.3 | 92.0 |
| FACT-TT | 82.3 | 89.7 | 91.6 | 78.0 | 85.4 | 87.6 | 92.9 | 94.5 | 81.5 | 89.1 | 90.5 | 94.6 | 96.3 | 83.7 | 91.3 | 91.4 | 94.7 | 96.0 | 84.1 | 91.6 |
| ARC | 82.9 | 89.4 | 93.2 | 78.5 | 86.0 | 88.2 | 92.6 | 95.6 | 82.5 | 89.7 | 90.6 | 94.3 | 95.5 | 83.9 | 91.1 | 91.5 | 94.7 | 95.7 | 85.5 | 91.9 |
| RLRR | 82.4 | 89.5 | 93.3 | 72.2 | 84.4 | 87.9 | 92.4 | 95.0 | 81.7 | 89.3 | 90.6 | 94.0 | 95.7 | 86.7 | 91.8 | 91.6 | 94.6 | 95.7 | 87.2 | 92.3 |
| APLA | 82.4 | 91.4 | 93.5 | 84.5 | 88.0 | 88.0 | 94.0 | 96.0 | 88.2 | 91.6 | 90.6 | 95.1 | 96.5 | 89.2 | 92.9 | 91.7 | 95.4 | 96.8 | 89.5 | 93.4 |

Table 10: *Applying other adaptation methods on $W_O$.* Competing methods are applied to $W_O$, isolating the effect of APLA's low-rank adaptation strategy. Notably, LoRA and ADAPTFORMER would be improved if they were placed at $W_O$ rather than their default locations.

| Method | Adaptation | Birds | Cars | AID | ISIC | Natrl. | Spec. | Struc. | Average |
|---|---|---|---|---|---|---|---|---|---|
| LoRA | Default | 87.5 | 93.4 | 95.4 | 86.5 | 83.4 | 86.5 | 63.1 | 78.2 |
| | On $W_O$ | 87.7 | 93.5 | 95.1 | 87.9 | 83.7 | 87.6 | 62.6 | 78.3 |
| ADAPTF. | Default | **88.4** | 93.1 | 95.4 | 85.6 | 84.0 | 87.2 | 59.8 | 77.3 |
| | On $W_O$ | **88.4** | 93.6 | 95.3 | 87.1 | 84.2 | 87.2 | 61.6 | 78.1 |
| FACT | Default | 87.8 | 93.0 | 95.4 | 85.1 | 84.7 | 87.4 | 64.5 | 79.1 |
| | On $W_O$ | 88.0 | 93.8 | 95.3 | 86.5 | 84.4 | 87.2 | 63.8 | 78.8 |
| **APLA** | On $W_O$ | 88.0 | **94.0** | **96.0** | **88.2** | **85.0** | **88.2** | 63.9 | **79.4** |

# B  ADDITIONAL EXPERIMENTAL RESULTS

## B.1  APPLYING OTHER ADAPTATION METHODS ON $W_O$

We evaluate LoRA Hu et al. (2021), FACT Jie & Deng (2023), and ADAPTFORMER Chen et al. (2022) on $W_O$ using ViT-B pretrained with DINOV2, and compare results across Birds, Cars, AID, ISIC, and VTAB-1k. Table 10 shows that adapting $W_O$ with these methods improves their performance compared to their original location, reinforcing the critical role of this $W_O$. APLA consistently outperforms them, highlighting the effectiveness of our simple low-rank strategy with random partial gradients.

## B.2  DETAILED RESULTS OF DIFFERENT MODEL SCALES

To examine if APLA scales well with model size, we utilize ViT models of varying sizes (ViT-S, ViT-B, ViT-L, and ViT-g), pre-trained with DINOV2 Oquab et al. (2023). Table 8 in the main text shows the average results of different adaptation methods across model scales, while Table 9 provides detailed per-dataset results. APLA appears to benefit from increased model capacity, performing exceptionally well with larger models. We further present a performance-efficiency trade-off comparison in terms of GPU memory consumption and latency during training across different model sizes in Figure 3. As the model size increases, APLA outperforms all other methods both in terms of predictive performance and costs during training.

## B.3  COMPUTATIONAL COSTS OF ADAPTATION METHODS

We analyze the computational costs of adaptation methods by measuring GPU memory footprint and latency during training and inference. Results are shown in Figure 4. During training, APLA is

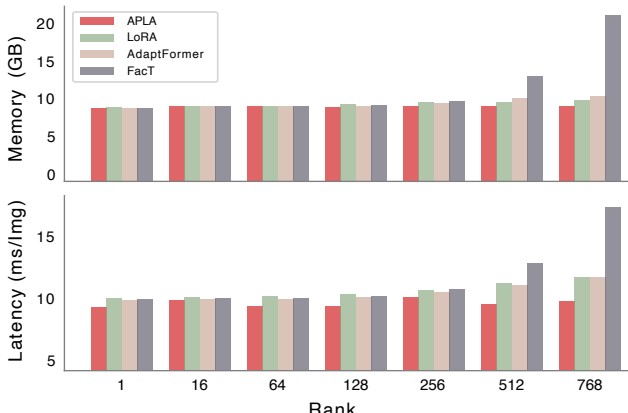

Figure 5: *Computational requirements of different adaptation methods during training for varying $r$ values.* For LoRA and FacT, $r$ denotes the rank, for APLA the number of tuned columns, and for AdaptFormer, the size of the bottleneck dimension.

the most efficient method in terms of GPU memory usage and latency and does not add any extra costs during inference. While APLA appears to tune more parameters than other adaptation methods, one should note that parameter count alone does not necessarily reflect the true computational costs. This inconsistency has been previously emphasized by other studies Dehghani et al. (2021); Cai et al. (2020)

### B.4 COMPUTATIONAL COSTS OF LOW-RANK ADAPTATION METHODS WHEN INCREASING $r$

In Table 10 in the main text, we investigated the impact of applying other low-rank adaptation methods on $W_O$, isolating the impact of the low-rank adaptation strategy. Using the same setup, here we analyze their computational costs with respect to the choice of rank $r$, considering GPU memory and latency during training. As shown in Figure 5, APLA is the only method that only minimally impacts memory and latency during, whereas other methods are affected to a larger extent as $r$ grows (*e.g.* FACT). Essentially, for any given rank $r$, APLA outperforms all other low-rank adaptation methods in terms of efficiency, requiring less GPU memory and enabling faster training. This advantage is due to APLA's more efficient low-rank strategy and its avoidance of introducing additional trainable parameters. This provides APLA a distinct advantage, allowing the rank to be freely adjusted for optimal results without any extra computational concerns.

### B.5 APLA ON INCREASING NUMBER OF VIT BLOCKS

We investigate the impact of applying APLA to increasing number of ViT blocks, when starting from the first layer and moving towards the last layers ("Bottom $\rightarrow$ Top") and the opposite direction ("Top $\rightarrow$ Bottom"), and present the results in Figure 6. Applying APLA to more blocks monotonically improves performance. As expected, applying APLA to later transformer blocks leads to greater performance improvements than applying it to the early ViT layers.

### B.6 THE EFFECT OF CHOICE OF $r$

In APLA, the hyperparameter $r$ is used to specify how many columns of the weight matrix $W_O$ are tuned. In Figure 7, we present the selected values of $r$ for general classification tasks (left), datasets with limited data (middle), and all datasets together (right).

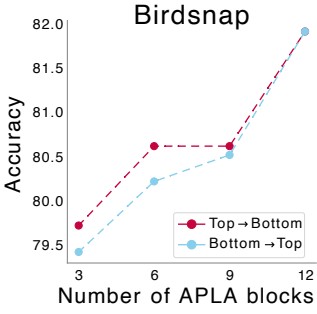 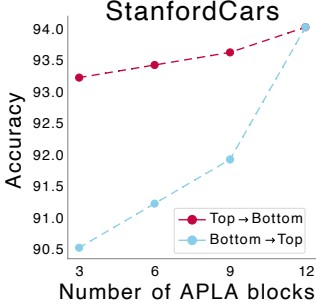

Figure 6: *Classification performance when applying APLA to an increasing number of attention blocks.* "Top-bottom" means applying APLA starting from the last ViT block and moving toward the first layer, while ""bottom-top" refers to applying it in the opposite direction.

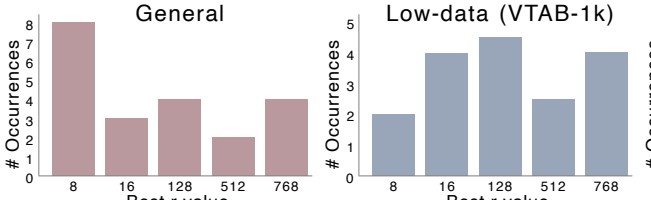

Figure 7: *Selected $r$ values across different datasets.* We report the optimal $r$ values, determined by grid searches, for general classification tasks (left), datasets with limited data (middle), and all datasets (right).

## B.7 Feature Visualization

As a last sanity check, we evaluate the quality of learned representations when using APLA and compare them with those obtained from other adaptation methods, similarly to Jia et al. (2022); Lian et al. (2022); Chen et al. (2022). In Figure 8 we use t-SNE Van der Maaten & Hinton (2008) to visualize the final representations derived from the [CLS] token of the last ViT block for various datasets from VTAB. Similar to other adaptation methods, APLA generates well-separated clusters for different classes, with data points from the same class positioned closely together.

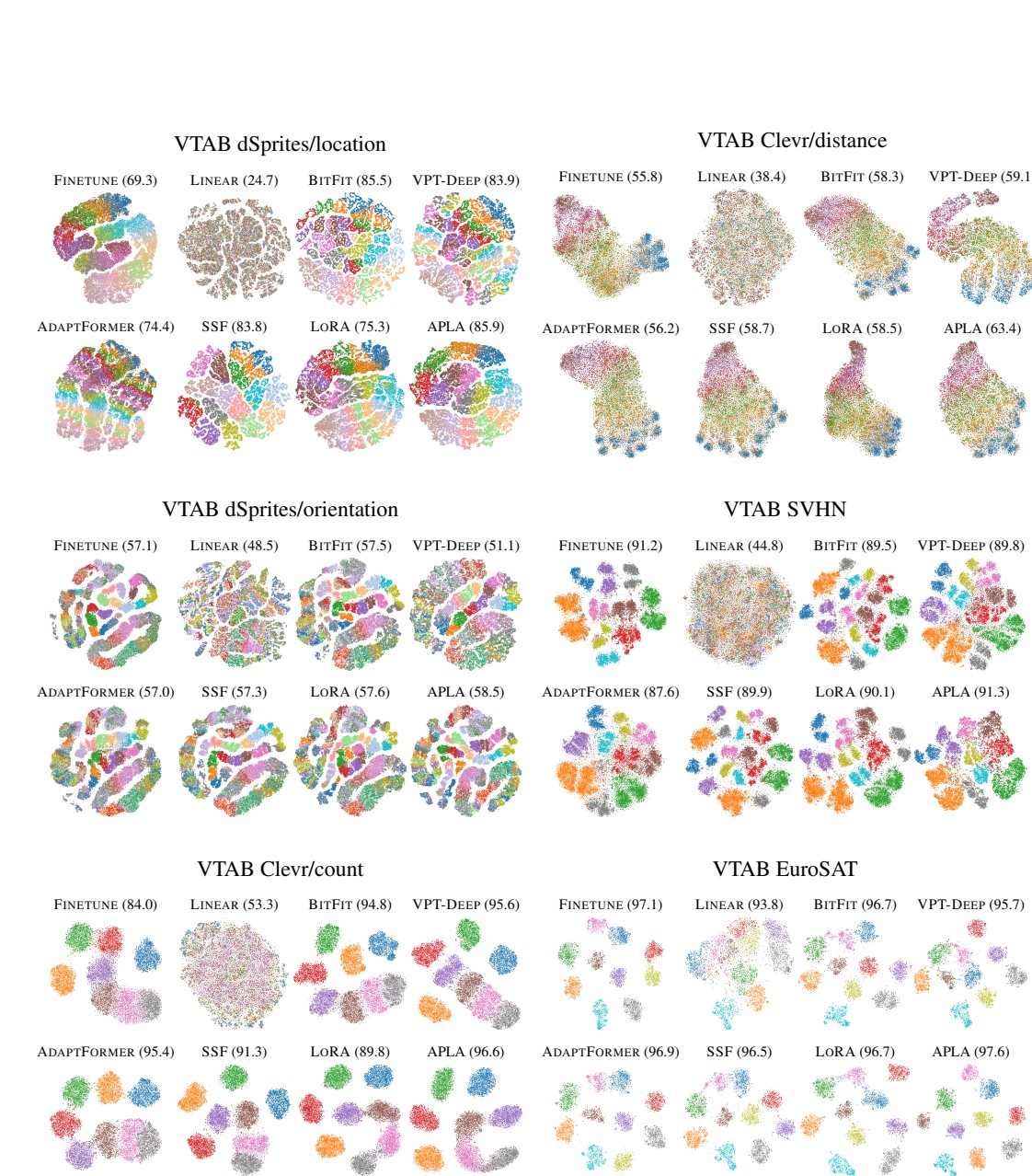

Figure 8: *t-SNE visualizations.* We plot the t-SNE visualizations of the output [CLS] embeddings on VTAB using ViT-B models pre-trained with DINOV2. All models have been adapted for each task. The numbers in parentheses indicate each adaptation method's classification performance for the task.

