# OpenReview forum: "APLA: A Simple Adaptation Method for Vision Transformers"
_ICLR.cc/2026/Conference — ICLR 2026 Conference Withdrawn Submission_

### Official Review · Reviewer_QQBy · 2025-10-30

**Soundness:** 2
**Presentation:** 3
**Contribution:** 2
**Rating:** 4
**Confidence:** 3

**Summary:**

This paper proposes Attention Projection Layer Adaptation (APLA), a method for efficiently adapting Vision Transformers (ViTs). The authors identify the attention output projection layer (WO) as the most critical component for adaptation. APLA achieves state-of-the-art performance by fine-tuning only this layer, or even just a randomly selected subset of its columns, without modifying the model architecture or introducing additional parameters. APLA outperforms full fine-tuning and 17 other leading adaptation methods, while reducing GPU memory usage by up to 52.63% and training time by up to 43.0%, with no inference overhead.

**Strengths:**

1. The method is simple and effective.
2. The experiments are thorough, with extensive testing conducted across various tasks and datasets.

**Weaknesses:**

1. The core of APLA lies in fine-tuning only a random subset of columns in the attention output projection layer (Wo). Although the experimental results are impressive, the method itself lacks theoretical depth or architectural innovation, resembling more of an "empirical observation" than a rigorous methodological breakthrough.
2. The number of columns fine-tuned in Wo, denoted as r, is a hyperparameter. As shown in Figure 7, the optimal r varies significantly across different datasets. Moreover, the authors claim to have used hyperparameter search to determine the best r. I argue that this may lead to reported metrics that merely reflect cherry-picked favorable outcomes from random trials, without sufficient evidence demonstrating the method’s genuine effectiveness.
3. The authors claim that applying other methods to Wo also yields improvements; however, as shown in Table 10, the gains are marginal at best, and the FACT method even shows a slight performance drop. This undermines the persuasiveness of the authors’ conclusions.
4. Prior studies[1] suggest that certain columns in Wo may correspond to core capabilities of the model, and fine-tuning these regions can lead to catastrophic forgetting. APLA lacks experiments addressing this critical issue.
[1] Jun Zhao, Zhihao Zhang, Yide Ma, Qi Zhang, Tao Gui, Luhui Gao, and Xuanjing Huang. 2023. Unveiling a core linguistic region in large language models.

**Questions:**

See the weaknesses.

---

### Official Review · Reviewer_t6ER · 2025-10-31

**Soundness:** 3
**Presentation:** 3
**Contribution:** 3
**Rating:** 6
**Confidence:** 4

**Summary:**

This paper introduces Attention Projection Layer Adaptation (APLA), a method for adapting Vision Transformers (ViTs) without altering the architecture or adding extra parameters. The authors identify the layer after the attention mechanism as key for adaptation and show that updating only this projection layer—or a subset of its weights—achieves state-of-the-art performance. APLA reduces GPU memory usage by up to 52.63% and training time by up to 43.0%, with no additional inference cost. In experiments across 46 datasets covering various tasks, APLA outperforms 17 other leading adaptation methods, including full fine-tuning, on classification, segmentation, and detection tasks.

**Strengths:**

1. The paper proposes a simple and effective fine-tuning method that outperforms 17 other baseline models across 46 datasets.

2. Fine-tuning only the attention projection layer is an interesting and potentially groundbreaking approach.

**Weaknesses:**

1. The findings are intriguing, but it is unclear whether similar results hold for larger models or models in other domains.

2. Since the method adapts the model by modifying its parameters, could there be issues with catastrophic forgetting or generalization?

3. Some works approach compression and adaptation from the token perspective, such as PYRA (pyra: parallel yielding re-activation for training-inference efficient task adaptation, ECCV 2024) and DyT (dynamic tuning towards parameter and inference efficiency for vit adaptation, NeurIPS 2024). A comparative analysis with these methods would be valuable.

4. The proposed method modifies the pre-trained model’s weights. Would it be feasible to add new randomly initialized weights to just a few columns of these weights? Theoretically, the results should be similar to the proposed method.

5. Why is fine-tuning only the attention projection layer more effective than other methods? What specific characteristics or role does this projection layer play? It would be helpful if the authors could provide more insights.

**Questions:**

N/A

---

### Official Review · Reviewer_k3Mr · 2025-10-31

**Soundness:** 3
**Presentation:** 3
**Contribution:** 2
**Rating:** 6
**Confidence:** 4

**Summary:**

This paper proposes APLA (Attention Projection Layer Adaptation), a parameter-selection approach for adapting Vision Transformers that tunes only a randomly chosen subset of columns in the attention output projection matrix $W_O$ in each block. The key empirical claim is that $W_O$ is the single most impactful ViT component to adapt. By simply tuning a random set of columns in $W_O$, APLA matches or even surpasses PEFT baselines and even full fine-tuning while lowering memory and training latency. Extensive experiments are conducted in this article to support the empirical claims, with evidence spans **46 datasets** including classification, VTAB-1k low-data, OOD robustness on ImageNet-A/R/C, ADE20K segmentation, and COCO detection/instance segmentation, and multiple backbones.

**Strengths:**

1. **Strong empirical support showing that APLA performs well across different downstream applications.** The author conducts extensive experiments in a wide scale, including (1) rich component studies proving that tuning only $W_O$ beats tuning other components and even full fine-tuning on average, (2) broad benchmarks showcasing that APLA achieves SOTA across over 40 datasets and across both coarse (classification) and dense (detection and segmentation) visual tasks, (3) OOD verification indicating that APLA exhibits OOD abilities.
2. **Efficiency.** Figure 3 and figure 4 in the appendix show that APLA incurs substantial GPU memory and training latency savings that scales with model size.
3. **Simplicity.** APLA is extremely simple and effective to deploy with no extra structures.

**Weaknesses:**

1. **Potential novelty issues.** Existing works have already shown that attention matrices are vital for fine-tuning transformer-structured models [1-2]. Additionally, selecting a subset of columns is also proven to be helpful [3]. APLA takes a step further, showing that selecting a subset of columns inside $W_O$ is the most effective approach, which appears incremental and limits the contributions claimed in L61-L75. Nevertheless, the empirical studies in this article are very valuable to follow.
2. **Cost for hyperparameter search.** Section A.1 in the appendix indicates that APLA demands an extensive searching process for the $r$ value. The authors should report the searching cost as it may be potentially computationally expensive.
3. **Stability.** Table 2 proves that randomly selecting  columns to tune yields the best performance. However, the number of the accuracy value is comparable with 3 other variants. The authors should further validate the "random" strategy by reporting the variance of each method and show that the randomly choosing columns does not affect stability.

[1] Three things everyone should know about Vision Transformers (ECCV 2022)

[2] Fine-Tuning Linear Layers Only Is a Simple yet Effective Way for Task Arithmetic (arXiv:2407.07089)

[3] RoCoFT: Efficient Finetuning of Large Language Models with Row-Column Updates (ACL 2025)

**Questions:**

See weaknesses.

---

### Official Review · Reviewer_T89s · 2025-11-05

**Soundness:** 2
**Presentation:** 2
**Contribution:** 3
**Rating:** 2
**Confidence:** 4

**Summary:**

This paper proposes an efficient training method for Vision Transformers (ViTs), claiming that using only a subset of the projection layer (W_O) parameters can achieve comparable or even better performance than full fine-tuning or adapter-based fine-tuning approaches, both in terms of accuracy and computational efficiency.

**Strengths:**

- **Comprehensive experimental evaluation**.
  - The paper presents an extensive set of experiments conducted across multiple datasets and settings, demonstrating strong empirical coverage.

- **Consistent performance gains**.
  -  The proposed method achieves consistent improvements over baselines across diverse configurations, suggesting its robustness and general applicability.

**Weaknesses:**

- **Insufficient methodological clarity**.
  - Sections 3.3 and 4 do not provide enough detail regarding the proposed method and experimental setup, particularly in terms of hyperparameter selection, sensitivity analysis, and implementation details. This lack of transparency makes reproduction of the results difficult.
  - See Questions 1–3 below.

- **Limited result interpretation**.
  - The paper does not clearly explain why random selection of projection layers should outperform the baselines. While Section 6 offers some discussion, a deeper, result-driven analysis would significantly strengthen the paper’s contribution and clarify the underlying mechanisms.
  - See Question 2.c below.

- **Readability and organization issues**.
  - The overall paper structure could be improved. Many figures and tables are presented in single-column format; using full-width (two-column) layouts would improve readability. In addition, improper citation formatting (e.g., using \citet instead of \citetp) affects readability and consistency.
  - While it’s understandable that summarizing numerous experiments is challenging, the main paper should highlight the most critical results and move less central details to the appendix for better focus and flow.

**Questions:**

- **Model adaptation and tuning**: These details are essential for understanding and reproducing the method.
  - When identifying which components to adapt, how was the model tuned?
  - If it was trained, what data, training objectives, and hyperparameters were used?

- **Random selection procedure**:
  - How many random trials were conducted? And how sensitive are the results to randomness?
  - Is the random selection applied across all transformer blocks and columns, or are all blocks adapted while only the columns within each block are randomly selected? Additionally, how many columns are ultimately selected per block?

- **Experimental details and efficiency**:
  - What hyperparameters were used for each dataset? How sensitive is the proposed method to hyperparameter choices? How are they selected?
  - How does the proposed approach’s efficiency compare to the time and computation required for hyperparameter search in conventional fine-tuning?
  - What is the actual computational saving when comparing full $W_O$ fine-tuning versus fine-tuning a randomly selected subset of $W_O$ parameters, including the search for the optimal random subset?

- **Results on ImageNet-C**:
  - Please provide detailed results for each corruption severity level to ensure a fairer and more transparent comparison.

---

### Official Review · Reviewer_dRV8 · 2025-11-09

**Soundness:** 3
**Presentation:** 3
**Contribution:** 2
**Rating:** 4
**Confidence:** 3

**Summary:**

This paper introduces APLA (Attention Projection Layer Adaptation), a parameter-efficient fine-tuning method for Vision Transformers. The key idea is to adapt only a randomly selected subset of columns in the projection layer following the multi-head attention mechanism, without adding new parameters or modifying the architecture. The authors validate APLA across 46 datasets and multiple tasks (classification, segmentation, detection), showing competitive or superior performance compared to 17 existing adaptation methods, including LoRA and AdaptFormer. APLA also demonstrates strong efficiency, reducing GPU memory usage by up to 52.6% and training time by 43%, with no inference overhead.

**Strengths:**

- Simplicity and practicality: The method is extremely easy to implement, requiring no architectural changes or additional parameters.
- Comprehensive evaluation: Experiments span diverse tasks, datasets, and model scales, including low-data and out-of-distribution scenarios.
- Efficiency gains: Significant reductions in memory and training time while maintaining or improving accuracy.
- Empirical finding: Identifying the projection layer (W_O) as the most impactful component for adaptation is a useful contribution.

**Weaknesses:**

- Limited novelty: The core idea—updating only part of an existing layer—is intuitive and could be considered incremental.
- Lack of theoretical explanation: The paper does not provide a strong intuition or analysis for why W_O is particularly effective for adaptation.
- Domain scope: Experiments focus exclusively on vision foundation models; applicability to other Transformer-based models (e.g., LLMs) is not explored.
- Missing comparisons: Several recent PEFT methods (e.g., FreqFit, DyT, ALaST, SNELL, Householder-based PEFT) are not included, which would strengthen the positioning of APLA.

[1] FreqFit: Enhancing Parameter-Efficient Fine-Tuning of Vision Transformers through Frequency-Based Adaptation, Arxiv 2024.

[2] ALaST: Adaptive Layer Selection for Efficient Vision Transformer Fine-Tuning, Arxiv 2024.

[3] DyT: Dynamic Tuning Towards Parameter and Inference Efficiency for ViT Adaptation, NeurIPS 2024.

[4] SNELL: Expanding Sparse Tuning for Low Memory Usage, NeurIPS 2024.

[5] Householder-based PEFT: Efficient Adaptation of Pre-trained Vision Transformer via Householder Transformation, NeurIPS 2024.

**Questions:**

1. Could APLA be applied to language models or multimodal Transformers? If so, what challenges or modifications would be needed?
2. Why does tuning W_O outperform full fine-tuning? Is this due to implicit regularization or reduced overfitting in low-data regimes?
3. Would combining APLA with other adaptation strategies (e.g., adapters or frequency-based modules) yield further improvements?
4. Can you include comparisons with very recent PEFT methods such as FreqFit, DyT, and Householder-based PEFT under common benchmarks?
5. lease include performance and memory usage comparisons with recent memory-efficient fine-tuning methods such as SNELL and its extensions, to better position APLA among the latest efficiency-oriented approaches.
6. Can a simple layer-level adaptation strategy like APLA remain competitive when foundation models such as DINOv3 already deliver strong zero-shot or frozen-backbone performance?

---

### Note · Authors · 2025-11-13

I have read and agree with the venue's withdrawal policy on behalf of myself and my co-authors.